# Interval Type-2 Neural Fuzzy Controller-Based Navigation of Cooperative Load-Carrying Mobile Robots in Unknown Environments

**DOI:** 10.3390/s18124181

**Published:** 2018-11-28

**Authors:** Chun-Hui Lin, Shyh-Hau Wang, Cheng-Jian Lin

**Affiliations:** 1Department of Computer Science & Information Engineering, Nation Cheng Kung University, Tainan 701, Taiwan; P78071044@mail.ncku.edu.tw (C.-H.L.); shyhhau@mail.ncku.edu.tw (S.-H.W.); 2Department of Computer Science & Information Engineering, National Chin-Yi University of Technology, Taichung 411, Taiwan

**Keywords:** evolutionary robot, navigation control, fuzzy control, wall-following control, cooperative carrying, interval type-2 neural fuzzy controller, artificial bee colony algorithm, grouping strategy

## Abstract

In this paper, a navigation method is proposed for cooperative load-carrying mobile robots. The behavior mode manager is used efficaciously in the navigation control method to switch between two behavior modes, wall-following mode (WFM) and goal-oriented mode (GOM), according to various environmental conditions. Additionally, an interval type-2 neural fuzzy controller based on dynamic group artificial bee colony (DGABC) is proposed in this paper. Reinforcement learning was used to develop the WFM adaptively. First, a single robot is trained to learn the WFM. Then, this control method is implemented for cooperative load-carrying mobile robots. In WFM learning, the proposed DGABC performs better than the original artificial bee colony algorithm and other improved algorithms. Furthermore, the results of cooperative load-carrying navigation control tests demonstrate that the proposed cooperative load-carrying method and the navigation method can enable the robots to carry the task item to the goal and complete the navigation mission efficiently.

## 1. Introduction

The robotics is rapidly progressed in recent years. Many researchers [1,2,3,4] have applied robots to various fields. Paul et al. [1] proposed a biomimetic robotic fish for informal science learning. Christopher et al. [2] presented a new robotic harvester that can autonomously harvest sweet pepper in protected cropping environments. Michail et al. [3] designed an autonomous robotic vehicle for monitoring the difficult fields to access or dangerous for humans. Maurizio et al. [4] developed effective emotion-based assistive behaviors for a socially assistive robot intended for natural human-robot interaction scenarios with explicit social and assistive task functionalities.

Several real-life tasks are easy for humans but difficult for robots. Robots do not possess brains to think as humans. Therefore, if people can make robots think like humans and judge things more flexibly, then robots can be used in various ways. Recent studies have evaluated robotic applications such as obstacle avoidance, path navigation, load carrying, and path planning.

In the real environment, the noise and interference cause sensor uncertainty. To solve complicated problems in engineering, we cannot successfully use classical methods because of various uncertainties typical for those problems. Several theories can be considered as mathematical tools for dealing with uncertainties. For a stochastically stable phenomenon in probability theory, it should exist a limit of the sample mean in a long series of trials. To test the existence of the limit, a large number of trials must be performed. Interval mathematics have arisen as a method of taking into account the errors of calculations by constructing an interval estimate for the exact solution of a problem. This is useful in many cases, but the methods of interval mathematics are not sufficiently adaptable for problems with different uncertainties. The most appropriate theory, for dealing with uncertainties is the fuzzy set theory [5] developed. Fuzzy set is defined by employing the fuzzy membership function. The fuzzy set in [5] is also called type-1 fuzzy set. The fuzzy set operations based on the arithmetic operations with membership functions. Rough set theory [6] is similar to fuzzy set theory, however the uncertain and imprecision in this approach is expressed by a boundary region of a set, and not by a partial membership as in fuzzy set theory. Rough set is defined by topological operations called approximations, thus this definition also requires advanced mathematical concepts. Molodtsov [7] did not impose only one way to set the membership function and used an adequate parametrization to soft set theory. In this study, because the fuzzy set theory is progressing rapidly at the present time, we focus on the fuzzy set for solving control problems. 

Some researchers have used the type-1 fuzzy system to robot control. Algabri et al. [8] proposed the fuzzy logic control system for controlling mobile robots and used evolutionary algorithm to adjust the parameters of a fuzzy controller. Lee et al. [9] adopted a fuzzy neural network based on reinforcement learning to implement the robot’s navigation control. Fathinezhad et al. [10] used supervised fuzzy sarsa learning (SFSL) to guide the robots’ navigation in environments that have obstacles. SFSL combines supervised learning and reinforcement learning to reduce learning and failure times. Infrared sensors were used in the study conducted by Anish [11]. These infrared sensors transmitted the distance between the robot and the obstacle to the adaptive network-based fuzzy inference system (ANFIS) controller. According to the aforementioned results, although the type-1 fuzzy system can successfully solve the navigation control and avoid collision problems, the control performance of robots is not good enough.

Recently, some researchers have extended the type-1 fuzzy set to the type-2 fuzzy set in a fuzzy system for solving robotic control [12,13,14,15], data classification [16], function approximation [17,18], and automatic control [19,20]. The type-2 fuzzy set incorporate uncertainty about the membership function into fuzzy set theory. Thus, type-2 fuzzy sets generalize type-1 fuzzy sets and more uncertainty can be handled. In [16,17,18,19,20], the experimental results have shown that the type-2 fuzzy set has a better performance than the type-1 fuzzy set for dealing with uncertainties. The output set corresponding to each rule of the type-2 fuzzy system is a type-2 fuzzy set. The type-reduction combines all these output sets and then performs a centroid calculation on this type-2 fuzzy set. The center-of-sets defuzzification method [21] is to replace each rule consequent set by a singleton situated at its centroid and then find the centroid of the type-1 set comprised of these singletons. However, general type-2 fuzzy systems are computationally intensive because type-reduction is very intensive. Things simplify a lot when secondary membership functions are set to interval sets. In this case, the secondary memberships are either zero or one. Therefore, the interval type-2 fuzzy set was proposed by [22,23] and adopted to reduce the computational complexity in this study. 

To resolve control problems, the backpropagation (BP) algorithm is used commonly to adjust the parameters of neural fuzzy controllers. However, the fast convergence ability of BP might cause the system to fall into local optimal solutions instead of global optimal solutions. Hence, evolutionary algorithms, such as the particle swarm optimization (PSO) [24], ant colony optimization [25], differential evolution (DE) [26], and artificial bee colony (ABC) [27], are proposed to find global optimal solutions. To speed up the convergence time without falling into local optimal solutions and improve accuracy, the group strategy is imported in the ABC algorithm. The proposed dynamic group artificial bee colony (DGBAC) balances the employed bees’ searching ability and changes the movement equations of employed bees and onlooker bees.

The goal of this paper proposes a cooperative load-carrying method of mobile robots to carry the task item and navigate in an unknown environment. The behavior mode manager is used efficaciously in the navigation control to switch between two behavior modes, wall-following mode (WFM) and goal-oriented mode (GOM), according to various environmental conditions. Additionally, an interval type-2 neural fuzzy controller (IT2NFC) based on DGABC is proposed in this paper. Reinforcement learning was used to develop the WFM adaptively. The proposed IT2NFC-based DGABC has several advantages, such as (1) Only small amount of parameters are required; (2) Interval type-2 fuzzy sets are used to reduce sensor-sensing noise and disturbance; (3) The effectiveness and robustness of the proposed controller are improved. Based on the aforementioned advantages, the behavior mode manager makes the moving path of the mobile robot smoother and automatically switches between the two modes to complete the task.

## 2. Structure of IT2NFC

In this section, the IT2NFC is introduced. The five layers in this structure are as follows: input layer, fuzzification layer, firing layer, output processing layer, and output layer. In Figure 1, X_1_~X_n_ represent the inputs of the IT2NFC, which are the distances between the nearby objects and the robot’s infrared sensors. $Y_{Left}$ and $Y_{Right}$ indicate the outputs of the IT2NFC, which are the speeds of the right and left wheels, respectively.

To improve the performance of the fuzzy system, the order reduction method, established by O. Castillo [19], was used in this paper. The reduction of order in the system can decrease the computational complexity. The rule in the IT2NFC is illustrated as follows:(1)$$\mathrm{Rule}\;j:\;\mathrm{IF}\;\;x_{1}\;\mathrm{is}\;{\tilde{A}}_{1}^{j}\;\mathrm{and}\;x_{2}\;\mathrm{is}\;{\tilde{A}}_{2}^{j}\ldots \;\mathrm{and}\;x_{n}\;\mathrm{is}\;{\tilde{A}}_{n}^{j}\;\mathrm{THEN}\;y_{Left}\;\mathrm{is}\;\omega_{Left}{}_{0}^{j}+\sum_{i=1}^{n}\omega_{Left}{}_{i}^{j}x_{i},\;y_{Right}\;\mathrm{is}\;\omega_{Right}{}_{0}^{j}+\sum_{i=1}^{n}\omega_{Right}{}_{i}^{j}x_{i}$$ where $x_{i}$ is the input variable, ${\tilde{A}}_{i}^{j}$ is an IT2FS, $y_{Left}$ and $y_{Right}$ are the output variables, $\omega_{0}^{j}+\sum_{i=1}^{n}\omega_{i}^{j}x_{i}$ is the output of the Takagi–Sugeno–Kang linear function, i represents the input from 1 to *n*, and j is the number of the rule.


1.Input Layer


In this layer, the input data is sent to the next layer directly without any computation.
(2)$$x_{i}=X_{i}$$


2.Fuzzification Layer


In this layer, named the membership function layer, the input data is calculated in a fuzzy manner. Each node is defined as an IT2FS. Figure 2 displays that each membership function is one Gaussian membership function and has an uncertainty mean $\left[m_{1},m_{2}\right]$ and a fixed standard deviation σ.
(3)$$u_{{\tilde{A}}_{i}^{j}}=\exp\left(-\frac{1}{2}\frac{{\left[x_{i}-m_{i}^{j}\right]}^{2}}{{\left(\sigma_{i}^{j}\right)}^{2}}\right)\equiv N\left(m_{i}^{j},\sigma_{i}^{j};x_{i}\right),m_{i}^{j}\in \left[m_{i1}^{j},m_{i2}^{j}\right]$$ where $m_{i}^{j}$ and $\sigma_{i}^{j}$ are the mean and deviation in the *i*th input in the *j*th Gaussian membership function. However, horizontal shifting causes the upper bound ${\bar{u}}_{\tilde{A}}$ and lower bound ${\underline{u}}_{\tilde{A}}$ of the membership function. The interval can be represented as $\left[{\bar{u}}_{{\tilde{A}}_{i}^{j}},\;{\underline{u}}_{{\tilde{A}}_{i}^{j}}\right]$.
(4)$${\bar{u}}_{{\tilde{A}}_{i}^{j}}=\{\begin{array}{ll} N\left(m_{i1}^{j},\;\sigma_{i}^{j};\;x_{i}\right), & x_{i}<m_{i1}^{j}\\ 1, & m_{i1}^{j}\leq x_{j}\leq m_{i2}^{j}\\ N\left(m_{i2}^{j},\;\sigma_{i}^{j};\;x_{i}\right), & x_{i}>m_{i2}^{j} \end{array}$$ and
(5)$${\underline{u}}_{{\tilde{A}}_{i}^{j}}=\{\begin{array}{c} N\left(m_{i2}^{j},\;\sigma_{i}^{j};\;x_{i}\right),\;\;\;\;\;\;\;\;x_{i}\leq \frac{m_{i1}^{j}+m_{i2}^{j}}{2}\\ N\left(m_{i1}^{j},\;\sigma_{i}^{j};\;x_{i}\right),\;\;\;\;\;\;\;\;x_{i}>\frac{m_{i1}^{j}+m_{i2}^{j}}{2} \end{array}$$



3.Firing Layer


In this layer, each node denotes the one rule node. Take the product of each fuzzy set and calculate the firing strength $F^{j}$. of each rule.
(6)$$F^{j}=\left[{\underline{f}}^{j},{\bar{f}}^{j}\right]$$
(7)$${\bar{f}}^{j}=\overset{n}{\underset{i}{\prod }}{\bar{u}}_{{\tilde{A}}_{i}^{j}}\;and\;{\underline{f}}^{j}=\overset{n}{\underset{i}{\prod }}{\underline{u}}_{{\tilde{A}}_{i}^{j}}$$ where ${\underline{f}}^{j}$ and ${\bar{f}}^{j}$ denote the upper and lower bounds of the firing strength.


4.Output Processing Layer


In this layer, the center-of-sets order reduction operation is used to produce IT1FS, and then the center-of-gravity method is used to de-fuzzy the set and obtain the output value $\left[y_{l},y_{r}\right]$.
(8)$$Y\left(x\right)=\left[y_{l},y_{r}\right]=\overset{\;}{\underset{a^{1}}{\int }}\cdots \overset{\;}{\underset{a^{M}}{\int }}\;\overset{\;}{\underset{f^{1}\in \left[{\underline{f}}^{1},{\bar{f}}^{1}\right]}{\int }}\cdots \overset{\;}{\underset{f^{M}\in \left[{\underline{f}}^{M},{\bar{f}}^{M}\right]}{\int }}\;\;\;\;\;\;1/\frac{\sum_{i=1}^{M}f^{i}a^{i}}{\sum_{i=1}^{M}f^{i}}$$

The output of the fuzzy set is defined as the following:(9)$$y_{l}=\frac{\sum_{j=1}^{M}{\underline{f}}^{j}\left(w_{0}^{j}+\sum_{i=1}^{n}w_{i}^{j}x_{i}\right)}{\sum_{j=1}^{M}{\underline{f}}^{j}}\;\;$$ and
(10)$$y_{r}=\frac{\sum_{j=1}^{M}{\bar{f}}^{j}\left(w_{0}^{j}+\sum_{i=1}^{n}w_{i}^{j}x_{i}\right)}{\sum_{j=1}^{M}{\bar{f}}^{j}}$$ where $y_{l}$ and $y_{r}$ indicate the upper and lower bound values of the output, $w_{i}^{j}$ represents the linked weights of each node, M represents the number of the rule, and n means the number of the input.


5.Output Layer


In this layer, the pervious output value is averaged to get the certain output from the neural network *y*.

(11)$$y=\frac{y_{l}+y_{r}}{2}$$

## 3. Structure of DGABC

### 3.1. Structure of ABC

The ABC, proposed by Karaboga in 2005, is a global optimization algorithm based on swarm intelligence. The advantages of ABC include fewer control parameters, parallel computing, global searching, simple computing, easy implementation, and robustness. The ABC consists of employed bees, onlooker bees, and scout bees. In the traditional algorithm, the employed bees search for a new food source within previous searching experiments and share the resource messages to onlooker bees. The onlooker bees wait for the messages from the employed bees and then search for the new food source according to employed bees’ information. The scout bees search all the food sources. If the food source gains do not improve, then the scout bees search a new food source in the search area randomly. The traditional ABC is introduced in the following steps [23].

Step 1: Initialization

Initialize the total number of food sources (FN), the maximum number of epochs (MEN), and the limit number of each non-improved food source (limit). Put each employed bee in the solution space randomly and set the location as a food source. The fitness of the food source is named the food source gains.
(12)$$x_{i}^{j}=x_{min}^{j}+rand\left[0,1\right]\left(x_{max}^{j}-x_{min}^{j}\right)$$ where $x_{i}^{j}$ is the initial value of the *i*th employed bee in the *j*th dimension, $x_{min}^{j}$ and $x_{max}^{j}$ are the minimum and maximum values of *j*th dimension in the search space, and the rand[0,1] is the random number from 0 to 1.

Step 2

Calculate the profits of the food source after each employed bee moves to a new food source.
(13)$$v_{i}^{j}=x_{ij}+rand\left[-1,1\right]\left(x_{i}^{j}-x_{k}^{j}\right)$$ where $v_{i}^{j}$ is the new location of the *i*th employed bee in the *j*th dimension, $x_{i}^{j}$ is the previous location of the *i*th employed bee in the *j*th dimension, $x_{k}^{j}$ is the location of the randomly chosen employed bee *k* in the *j*th dimension, and the rand[−1,1] is a random number from −1 to 1. 

Step 3

Use roulette wheel selection (Equation (14)) to determine which food source should be searched by each onlooker bee. The onlooker bee searches for the nearby food source and then calculates the profits of that food source.
(14)$$P_{i}=\frac{fit_{i}}{\sum_{n=1}^{SN}fit_{n}}$$ where $P_{i}$ is the probability of the *i*th food source being selected, $fit_{i}$ is the profits of the *i*th food source, and FN is the number of the food source.

Step 4

If the food source gains does not improve after limit times, then abandon this food source. According to Equation (12), the scout bee will find a new food source and replace the abandoned food source. 

Step 5

Record the highest food source gains and set it as the best solution in the algorithm. 

Step 6

Recursively run the algorithm until the current epoch is equal to MEN. If YES, then stop the algorithm and output the best solution. If NO, then return to step 2. 

### 3.2. Structure of DGABC

The proposed DGABC is used to adjust the parameters in the IT2NFC. DGABC balances the search ability of the traditional ABC and avoids the early converge problem. The proposed DGABC uses the dynamic grouping strategy and reforms the movement equations of the employed and onlooker bees to improve the convergence speed and the algorithm’s performance. The evolution flowchart is displayed in Figure 3.

In DGABC, each bee represents an IT2NFC and a set of parameters. Figure 4 depicts the coding of all the parameters including the uncertain mean $\left[m_{i1}^{j},m_{i2}^{j}\right]$, standard deviation $\sigma_{i}^{j}$, and link weights $w_{Left}{}_{0}^{j},w_{Left}{}_{i}^{j},w_{Right}{}_{0}^{j},w_{Right}{}_{i}^{j}$.

In the traditional ABC, the number of employed and onlooker bees is equal. However, improper setting of the number of employed and onlooker bees might result in the unequal searching ability of global and local searching and lower the performance.

The group strategy is used to bring parity to the searching competence. Bees are separated into groups dynamically according to their performance. The best bee is selected as the leader. To maintain consistency with the traditional ABC, in which employed bees lead onlooker bees, the leader in each group is selected as an employed bee and the rest of the bees in the group are selected as onlooker bees. During training, the proportion of two types of bees can be adjusted automatically. 

Using similarity evaluation to categorize individuals can prevent the following three cases (depicted in Figure 5).

Case 1:

$B_{1}$ and $B_{2}$ are in a similar location. However, the fitness values are very different. 

Case 2:

$B_{3}$ and $B_{4}$ have similar fitness values. However, the distance between them is large.

Case 3: 

$B_{5}$ and $B_{6}$ have similar fitness and are located close to each other. Based on similarity evaluation, and $B_{6}$ are similar individuals.

Some researchers [20] have used thresholds to group the individuals. However, the fixed threshold results in uneven number of groups in early and late epochs. Therefore, dynamic adjustment of the threshold is proposed in this paper. 

The DGABC steps are as follows (depicted in Figure 6):Dynamic Grouping

Step 1: Sorting

Sort the bees from the best to worst and initiate 0 to all the groups.

Step 2: Calculate the threshold value of similarity.

Find the fittest bee among the ungrouped bees. Then, select the fittest bee as the new group leader, name the group *g*, and calculate the fitness threshold.
(15)$$DIS^{g}=\sum_{i=1}^{SN}\sum_{j=1}^{D}\sqrt{{\left(Leader_{j}^{g}-Bee_{j}^{i}\right)}^{2}},\;\;\;if\;Bee^{i}\;is\;ungrouped$$
(16)$$FIT^{g}=\sum_{i=1}^{SN}\left|Fit\left(Leader^{g}\right)-Fit\left(Bee^{i}\right)\right|\;,\;\;\;if\;Bee^{i}\;is\;ungrouped$$
(17)$$Average\_Distance\left(ADIS^{g}\right)=\frac{DIS^{g}}{NC}$$
(18)$$Average\_Fitness\left(AFIT^{g}\right)=\frac{FIT^{g}}{NC}$$ where $ADIS^{g}$ and $AFIT^{g}$ are the distance and fitness thresholds, respectively, in group g, $Leader_{j}^{g}$ is the leader location in group g, $Fit\left(Leader^{g}\right)$ is the fitness value of leader in group *g*, SN is the total number of bees, is the dimension, and NC is the total number of bees in group 0. 

Step 3: Evaluate the similarity

Calculate the distance $\left(Dis^{i}\right)$ and fitness $Fit^{i}$ between ungrouped bee $Bee^{i}$ and the leader.
(19)$$Dis^{i}=\;\sum_{j=1}^{D}\sqrt{{\left(Leader_{j}^{g}-Bee_{j}^{i}\right)}^{2}}$$
(20)$$Fit^{i}=\left|Fit\left(Leader^{g}\right)-Fit\left(Bee^{i}\right)\right|$$

If $Dis^{i}<ADIS^{g}$ and $Fit^{i}<AFIT^{g}$, then the bee $Bee^{i}$ is similar to the group leader. Place them into the same group and change the group number to *g*. 

Step 4: Checking

Check whether there are ungrouped bees. If YES, then go back to step 1 and select the fittest bee in the ungrouped bees as the new group leader and repeat step 1 to step 3. If NO, all the bees are grouped and grouping is complete. 

2.Employed Bees Phase

In the traditional ABC, employed bees randomly select their food source searching direction. In the proposed method, the searching direction is improved. The global best solution is used in the movement equation to enable the employed bees to move to the better solution. Moreover, the proposed method maintains the random search mechanism.
(21)$$v_{j}^{i}=x_{j}^{i}+\emptyset 1_{j}^{i}\left(x_{j}^{i}-x_{j}^{best}\right)+\emptyset 2_{j}^{i}\left(x_{j}^{i}-x_{j}^{k}\right)$$ where $x_{j}^{best}$ is the best solution in the group and $\emptyset 1_{j}^{i}$ and $\emptyset 2_{j}^{i}$ are random numbers between [−1,1].

3.Onlooker Bees Phase

The grouping strategy alters the traditional onlooker bees’ searching method to the leader (employed bees). The assigned leader must guide the teammates (onlooker bees) to search for a food source.
(22)$$v_{j}^{i}=x_{j}^{i}+\emptyset_{j}^{i}\left(x_{j}^{i}-x_{j}^{Leader}\right)$$ where $\emptyset_{j}^{i}$ is a random number between [−1, 1] and $x_{j}^{Leader}$ is the location of the leader in the group.

## 4. Control of the Mobile Robot

In this paragraph, the mobile robot is introduced first and then the method of wall-following control is explained in the following section.

### 4.1. Description of the Mobile Robot

E-puck mobile robots, new small-sized mobile robots manufactured by EPFL, were used in this research. As displayed in Figure 7a, the robot was controlled by the DSPIC processor. Several standard configurations such as a proximal infrared sensor, voice sensor, accelerometer, and camera are included in the robot. The mobile robots connect to each other through Bluetooth and use Bluetooth to communicate with the computers. Nowadays, e-puck mobile robots are widely used. Applications for e-buck include mobile mechanical engineering, real-time programs, interpolation systems, signal transmission, image transmission, combination of sounds and images, human–computer interaction, and robot internal communication.

Figure 7b shows that e-puck is a two-wheeled mobile robot with eight 360° surrounding infrared sensors from S_0_–S_7_. The diameter of e-puck is 7 cm, height is 5 cm, and maximum speed is 15 cm/s. The maximum distance between the mobile robot and nearby objects is 6 cm, and the minimum distance is 1 cm. 

### 4.2. Wall-Following Control of the Mobile Robot

In this study, RL was used to implement the wall-following behavior. The robot can evaluate the performance by defining the suitable fitness functions in the training environment even when predefined rules and pre-specified training data are not provided. The wall-following control learning flowchart is displayed in Figure 8.

The IT2NFC controller has four input signals (S_0_, S_1_, S_2_, and S_3_) and two output signals (V_L_ and V_R_). The input signal S_i_ denotes the distance from the *i*th infrared sensor and the output signals represent the turning speeds of the left and right wheels.

Figure 9 exhibits the training environment. The goal is to enable the mobile robot to adapt to any type of unknown environments including straight lines, sharp corners, obtuse corners, smooth corners, and U-curve corners.

To prevent the robot from moving far away from the walls or colliding with the obstacles, three terminate actions were designed in this study.

Action 1:

The total moving distance must be more than the preset maximum distance (the distance of navigating the training environment), which means that the robot successfully detours the training environment once.

Action 2:

As Figure 10a indicates, the detected distance from one of the sensors is less than 1 cm, which means that the robot hits the wall. 

Action 3:

As Figure 10b illustrates, the detected distance from one of the side sensors is more than 6 cm, which means that the robot is deviating from the wall.

To evaluate the performance during the learning, the fitness function is proposed in this paper. The fitness function has three sub-fitness functions, namely the total distance of the robot, the average distance between the robot and the wall, and the parallel degree between the robot and the wall (*SF*_1_, *SF*_2_, and *SF*_3_). 

SF_1_

If the robot’s moving distance $R_{dis}$ is close to the preset distance $R_{total}$, then the robot is close to detouring the training environment.
(23)$$SF_{1}=R_{total}-R_{dis}$$

If the robot’s moving distance is more than the preset distance $R_{total}$, then the robot traverses the training environment successfully and $SF_{1}=0$.

2.SF_2_

To ensure the robot and the wall stay at a constant distance, the distance WD(t) between the robot’s side sensor $S_{2}\left(t\right)$ and the wall is calculated in every time step, as indicated in Figure 11a.
(24)$$WD\left(t\right)=\left|S_{2}\left(t\right)-d_{wall}\right|$$
(25)$$SF_{2}=\frac{\sum_{t=1}^{T_{stop}}WD\left(t\right)}{T_{stop}}$$ where $d_{wall}$ is the expected distance between the wall and $T_{stop}$ is time step. In this study, $d_{wall}$ was selected as 4 cm.

If the robot stays at a constant distance with the wall, then $SF_{2}=0$.

3.SF_3_

To ensure the robot remains parallel with the wall, the law of cosines is used on the distances from the robot’s front right sensor and the side sensor to calculate the length x(t). Like Figure 11b.
(26)$$RS_{1}=r+S_{1},RS_{2}=r+S_{2}$$
(27)$$x\left(t\right)=\sqrt{RS_{1}{}^{2}+RS_{2}{}^{2}-2RS_{1}RS_{2}\cos(45{}^{\circ})}$$ where *r* is the radius of the robot, $RS_{1}$ is the distance from the sensor $S_{1}$, and $RS_{2}$ is the distance from the sensor $S_{2}$. If the side sensor $S_{2}$ is vertical to the wall, then x(t) is equal to and $SF_{3}=0$.
(28)$$SF_{3}=\frac{\sum_{t=1}^{T_{stop}}\left|RS_{2}-x\left(t\right)\right|}{T_{stop}}$$

Therefore, the fitness function $\bar{F}$ can be defined as the following:(29)$$\bar{F}=\frac{1}{1+\left(SF_{1}+SF_{2}+SF_{3}\right)}$$

## 5. Cooperative Load-Carrying and Navigation Control of Mobile Robots

In this section, the cooperative load-carrying control of multi-mobile robots is depicted. First, set two mobile robots in the same direction and place the load item upon them. The front robot is the leader and the other is the follower. The distance between the two robots is 15 cm. Figure 12 demonstrates that the leader explores the environment and guides the follower. The follower assists the leader in carrying the item.

### 5.1. Cooperative Load-Carrying Wall-Following Control of Mobile Robots

The wall-following control method is used for the leader and the follower. The auxiliary controller is attached to the follower. Figure 13 presents the flowchart of cooperative load-carrying wall-following control. The auxiliary controller has five inputs, which consist of the distances detected from the four sensors on the follower $\left(S_{0},\;S_{1},\;S_{2},\mathrm{and}\;S_{3}\right)$ and the distance between the leader (*R_d_*). The two outputs (*V_L_* and *V_R_*) are the speeds from left and right wheels, respectively. Figure 14 exhibits the training environment that includes straight lines, sharp corners, obtuse corners, smooth corners, and U-curve corners.

In this control method, the leader only follows the wall and the follower assists the leader to carry the item and adjusts its own output from the auxiliary controller to match the leader’s output. To prevent the follower and leader from staying at an unsafe distance or failing the mission, the following five terminations are designed in this study.

Case 1

If one of the sensors on the follower reaches a distance less than 1 cm, then the follower hits the wall. 

Case 2

If the follower’s side sensor reaches a distance more than 6 cm, then the follower deviates from the wall. 

Case 3

If Rd is less than 10 cm or more than 20 cm, then the leader and the follower are too close to each other or too far away from each other, respectively. Improper value of Rd might cause the leader to make a wrong detection or the item to fall. 

Case 4

If the sensor under the item is smaller than the robot’s height $\left(R_{h}\right)$, then the task fails. 

Case 5

If the side sensor detects a distance less than 1 cm or more than 7.5 cm, then the item is too close to the wall or too far away from the wall.

In practical testing experiments, the side and bottom sensors are removed. When the robots encounter a termination case, the fitness value F(·) is calculated by the robot’s time steps. Then, the robots return to the initial point and restart the program again until they traverse the training environment successfully.
(30)$$F\left(\cdot \right)=T_{stop}/5000$$

A maximum time step of 5000 is allocated in this study. The longer the robots move, the higher the fitness value gets.

### 5.2. Cooperative Load-Carrying Navigation Control of Mobile Robots

Known and unknown environments are the two types of environments in existence. A navigation control method to effectively assist mobile robots to navigate unknown environments is proposed in this paper. The behavior mode manager was used to switch between the goal-oriented mode (GOM) and the wall-following mode (WFM). The robots were initialized in the GOM on detecting the obstacles, then switched to the WFM. Otherwise, the robots remain in the GOM.

GOM

In an unknown environment, the goal position is the only information available to the robots. Therefore, Figure 15 shows an angle $\theta_{RG}$ and the robot moves toward the goal.
(31)$$\theta_{RG}=\theta_{Robot}-\theta_{Goal}$$ where $\theta_{Robot}$ is the angle between the robot and the axis *x* and $\theta_{Goal}$ is the angle between the goal and the axis *x*. 

Figure 16, in the GOM, the follower moves in the same direction and with the same speed with the leader. 

Behavior mode manager

Figure 17 shows the three areas of the robot $O_{1}$, $O_{2}$, and $O_{3}$. Depending on which side of the sensors detect the goal $\left(O_{i}\right)$ or which sensors detect the obstacles $\left(S_{i}\right)$, the behavior mode manager determines which side of the wall the robots should follow. 

Left-side wall-following:The goal is in $O_{1}$ and $S_{7}$ or $S_{6}$ senses the obstacles.The goal is in $O_{2}$ and $S_{7}$, $S_{6}$, $\mathrm{and}\;S_{5}$ sense the obstacles. 

Right-side wall-following:The goal is in $O_{1}$ and $S_{0}$ or $S_{1}$ senses the obstacles.The goal is in $O_{3}$ and $S_{0},S_{1}$, and $S_{2}$ sense the obstacles.

Prior to the robot’s switch to the WFM, the behavior mode manager deduces which robot detects the obstacles first. Figure 18 displays that if the leader detects the obstacles, then the leader will pre-rotate first to stay parallel to the wall. The leader then sends $\theta_{L}$ to the follower, and the follower rotates $\pi -\theta_{L}$ degrees. Once both robots are within a safe distance and parallel to the wall, the WFM is executed. After they pass the obstacles, the GOM switches back again. Figure 19 analyses the actions of the follower. 

Figure 20 reveals that if the follower detects the obstacle, then both the robots switch to the WFM directly.

As shown in Figure 21, during the WFM, if the obstacle is in the follower’s *O*_1_ area and the right-side sensor $S_{1}$ detects a distance larger than 6 cm (if the robots are in the left-side WFM, then the distance is detected by the left-side sensor $S_{6}$), then the item on the robots is past the obstacle. The behavior mode manager can switch the WFM back to the GOM. 

## 6. Experimental Results

Simulation is very important in the robot design process, and the developed algorithm can be quickly verified. At the same time, for robot learners, simulation tools can greatly reduce the cost of learning. The robot simulation platform integrates a physics engine which can calculate motion, rotation and collision according to the physical properties of the object. The simulation tools commonly used in robots are Gazebo, V-REP, and Webot. Table 1 shows the features of various robot simulator software. In Table 1, the V-Rep and Webots support Linux, mac OS, and Windows development environments, whereas the Gazebo only supports Linux development environment. The V-Rep simulation software has less functional support than Gazebo and Webots. Therefore, in this study Webot is used as a robot simulation platform to verify the proposed IT2NFC controller based on DGABC learning algorithm and perform the WFM control and cooperative load-carrying of mobile robots.

### 6.1. Results of WFM Control

The performance and stabilization for DGABC were compared with those for other algorithms. Best, worst, average, standard deviation (STD), and the number of successful runs after ten runs and the average executing times are mentioned in Table 2. The number of successful runs demonstrates that the mobile robots can traverse the training environment one time successfully from the start point to the goal during the simulation. And the learning curves of each algorithms are shown in Figure 22. Table 2 and Figure 23 are under the same conditions. In Table 2, the proposed method exhibited a higher learning effect than improved ABC, ABC, and DE. In addition, the average executing time of proposed method is shorter than those of other methods. 

In addition, we have also compared the proposed IT2NFC (type-2 fuzzy sets) based on DGABC with the IT1NFC (type-1 fuzzy sets) based on DGABC. Table 3 shows the performance comparison of IT1NFC and IT2NFC. The experimental results show the performance of IT2NFC (type-2 fuzzy sets) is better than the IT1NFC (type-1 fuzzy sets)” 

### 6.2. Results of Cooperative Load-Carrying WFM Control

The results revealed that the trained control method was implemented for cooperative load-carrying mobile robots. Table 4 compares the performance of the proposed WFM with that of other algorithms in the simulation. The better performance and higher stabilization for cooperative load-carrying WFM control in DGABC is demonstrated in Figure 24. Figure 25 shows the moving path of two mobile robots in the training environment. The PSO did not display any advantage in cooperative load-carrying WFM control. Therefore, the proposed DGABC method exhibited better performance in this control.

### 6.3. Results of Cooperative Load-Carrying Navigation Control

Implementation of the proposed method in cooperative load-carrying mobile robots in unknown environments is demonstrated in Figure 26. In this experiment, the average distance between two mobile robots (RAD) and the average distance between the wall and the follower robot (FAD) were the two critical standards for evaluating the performance of the proposed method. As shown in Table 5, if the RAD value is too large, the two mobile robots do not stay within the safe distance, which might cause the object to fall. On the other hand, if the FAD value is too large or too small, then the WFM does not perform satisfactorily when the mobile robots are turning around the corner. This might cause the object to shift during the mission, the object to fall, and thus, the mission to fail.

### 6.4. Discussion

In this study, we have successfully implemented the cooperative load-carrying task of two mobile robots by using the proposed IT2NFC-based DGABC. Next, the proposed method will be extended to three and more mobile robots for the cooperative load-carrying task. For example, Figure 27 shows the cooperative load-carrying task of three mobile robots. First, the wall-following control method of the cooperative load-carrying task is used for the leader, the follower-1, and the follower-2. Second, the auxiliary controller is attached to the followers based on the description in Section 5.1. Finally, the behavior mode manager in navigation control is also used to switch between the GOM and the WFM in Section 5.2. Therefore, the cooperative load-carrying task of three mobile robots would be implemented. According to the aforementioned steps, the proposed method can be successfully extended to the cooperative load-carrying task of three and more mobile robots.

## 7. Conclusions

The proposed IT2NFC-based DGABC improves the search capacity and shortens the convergence speed for avoiding falling into local optimal solutions. Using the proposed method, the mobile robots can develop the controller adaptively because DGABC does not need any predefined rule set nor does it require pre-specified training data. According to the environmental situations, mobile robots use the behavior mode manager to switch between WFM and GOM. Additionally, the pre-rotate mechanism ensures the follower robot follows the wall perfectly and allows the leader robot to complete the cooperative load-carrying task in unknown environments.

The cooperative load-carrying task is so complex for the robots that several factors, such as the robot speed, the object payload, and the working environment, should be taken into consideration. Therefore, we focus on implementing the cooperative load-carrying task of two mobile robots in this study. In the future work, we will consider implementing the cooperative load-carrying task of two and more real mobile robots in the unknown environments.

## Figures and Tables

**Figure 1 sensors-18-04181-f001:**
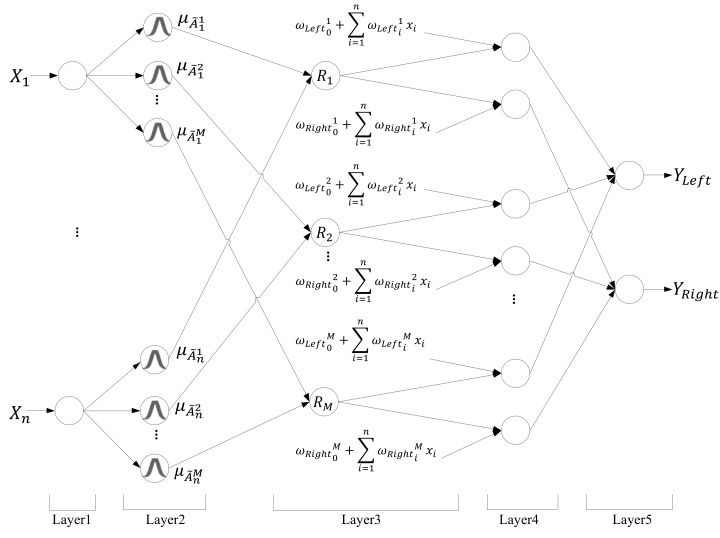
Structure of IT2NFC.

**Figure 2 sensors-18-04181-f002:**
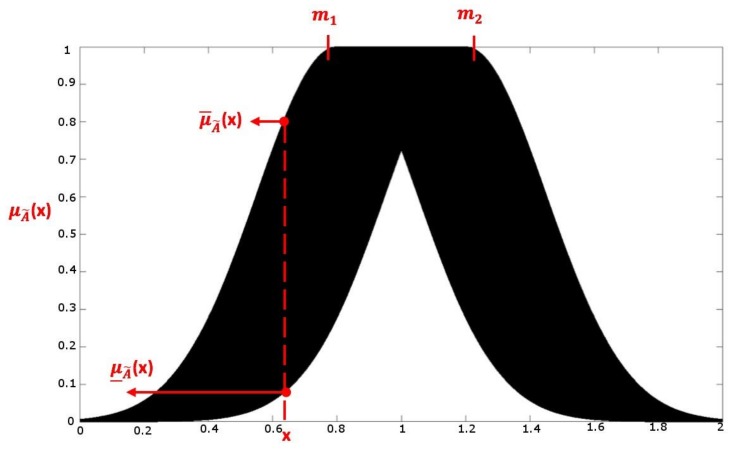
Interval type-2 fuzzy sets.

**Figure 3 sensors-18-04181-f003:**
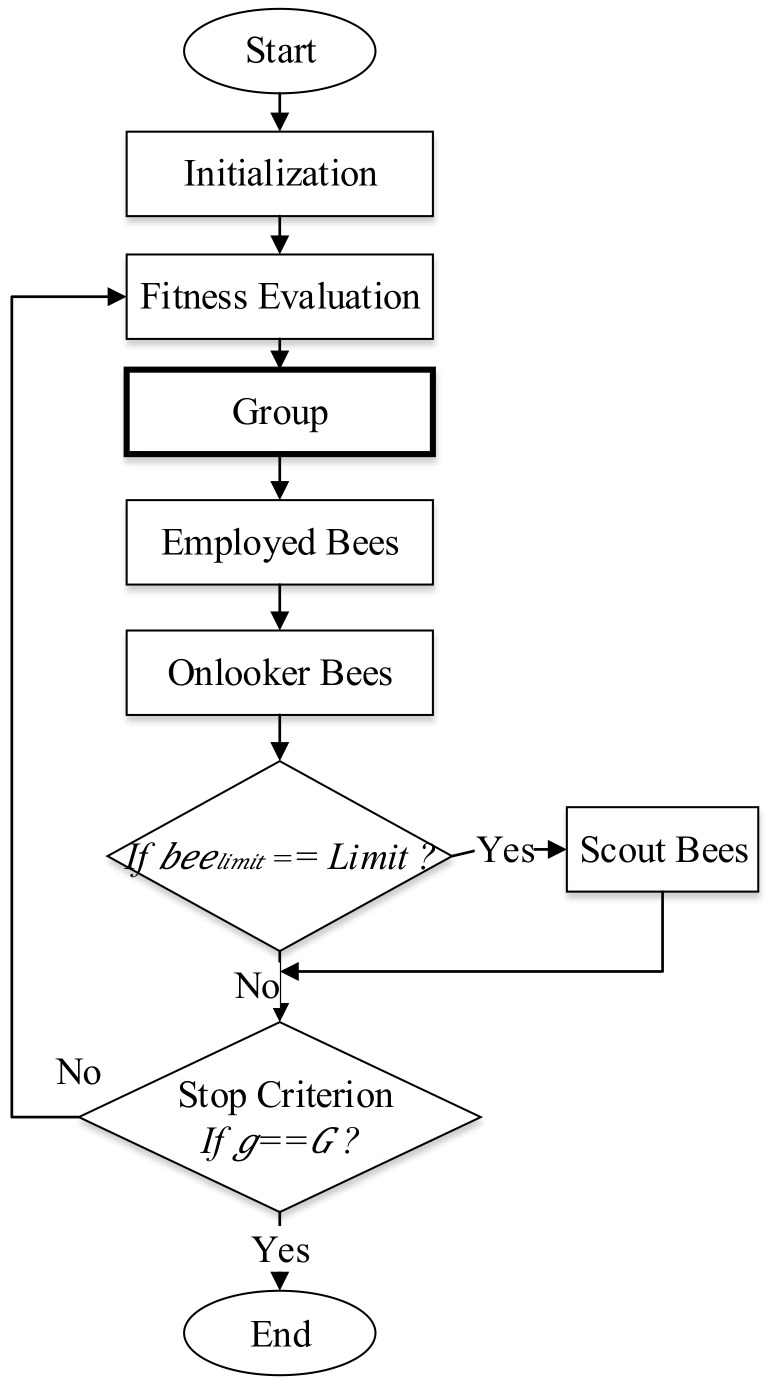
Flowchart of DGABC.

**Figure 4 sensors-18-04181-f004:**
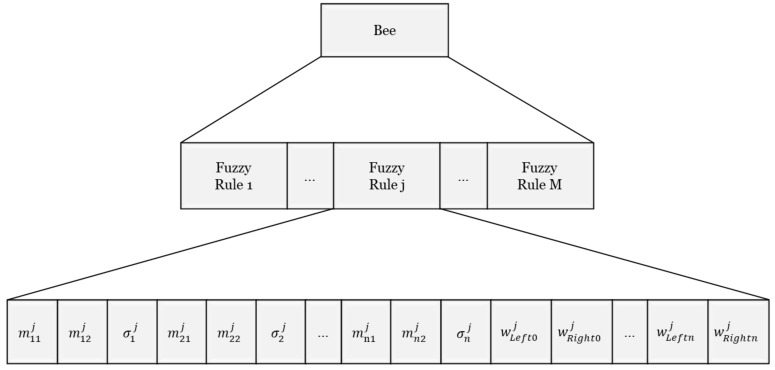
Coding of a bee.

**Figure 5 sensors-18-04181-f005:**
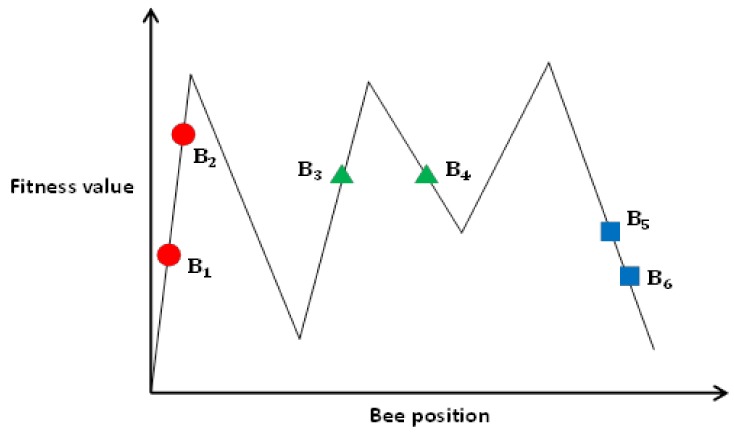
Similarity evaluation of individuals.

**Figure 6 sensors-18-04181-f006:**
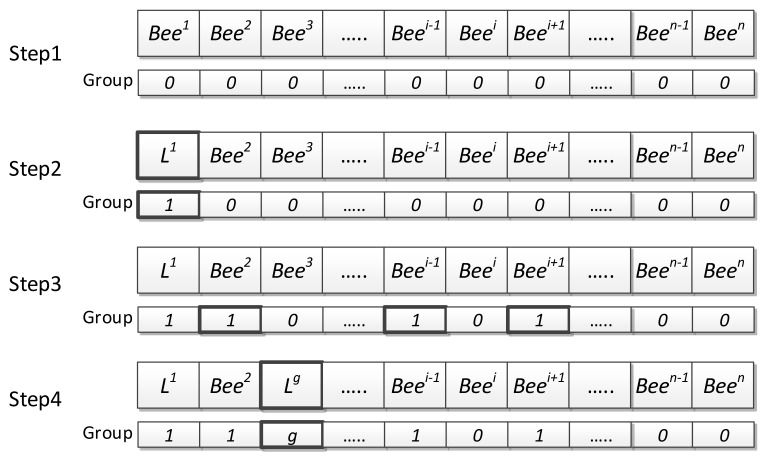
Dynamic grouping method.

**Figure 7 sensors-18-04181-f007:**
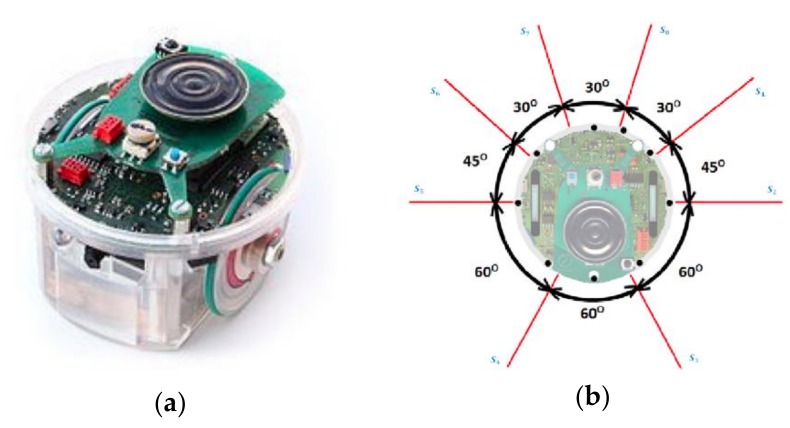
(**a**) E-puck mobile robot; (**b**) E-puck 360° infrared sensors.

**Figure 8 sensors-18-04181-f008:**
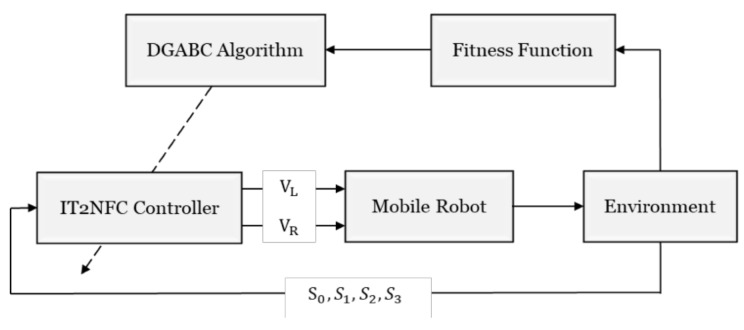
Flowchart of wall-following control learning.

**Figure 9 sensors-18-04181-f009:**
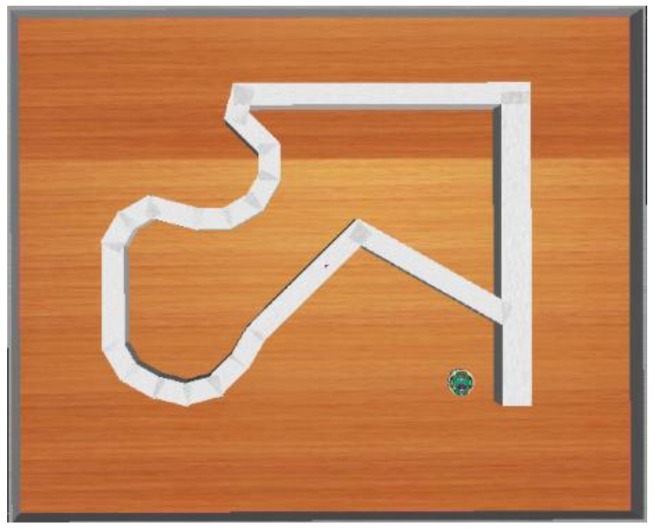
Training environment.

**Figure 10 sensors-18-04181-f010:**
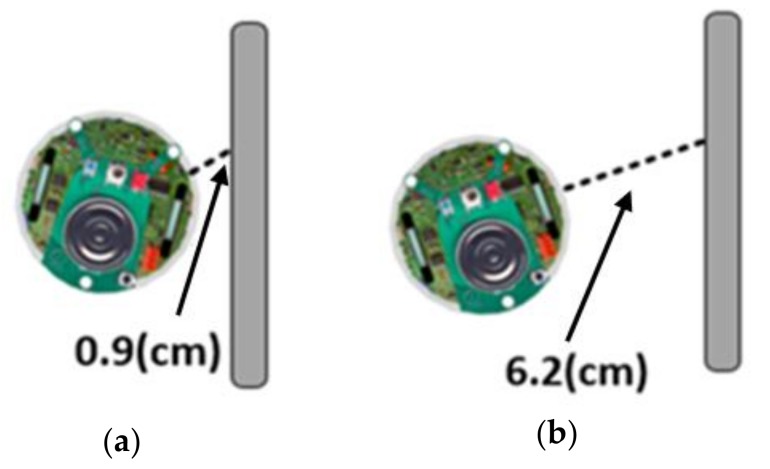
(**a**) Robot hits the wall; (**b**) Robot deviates from the wall.

**Figure 11 sensors-18-04181-f011:**
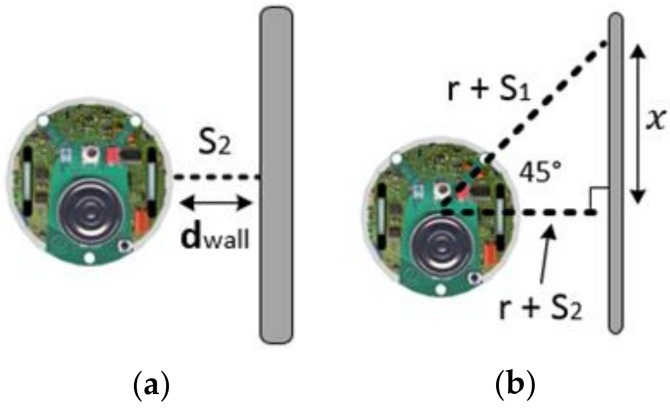
(**a**) Average distance; (**b**) Parallel degree.

**Figure 12 sensors-18-04181-f012:**
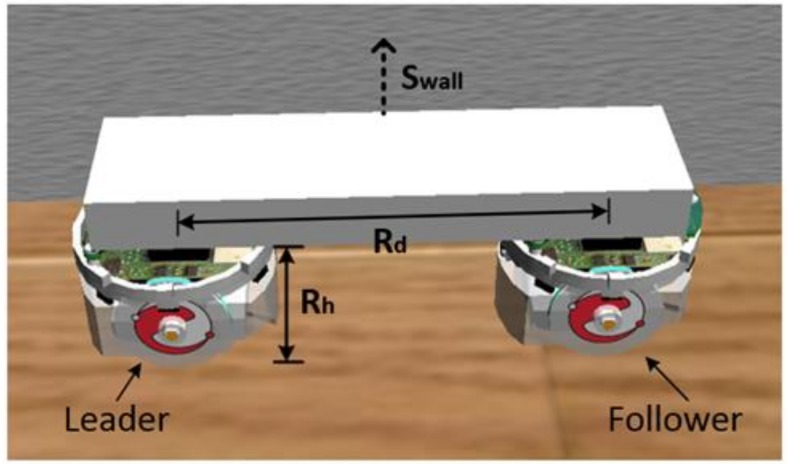
Cooperative load-carrying of two robots.

**Figure 13 sensors-18-04181-f013:**
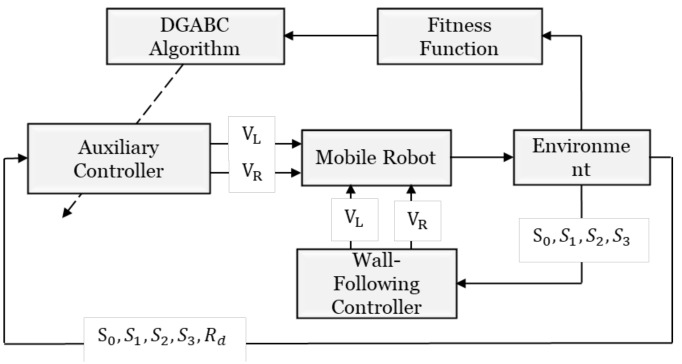
Cooperative load-carrying wall-following control.

**Figure 14 sensors-18-04181-f014:**
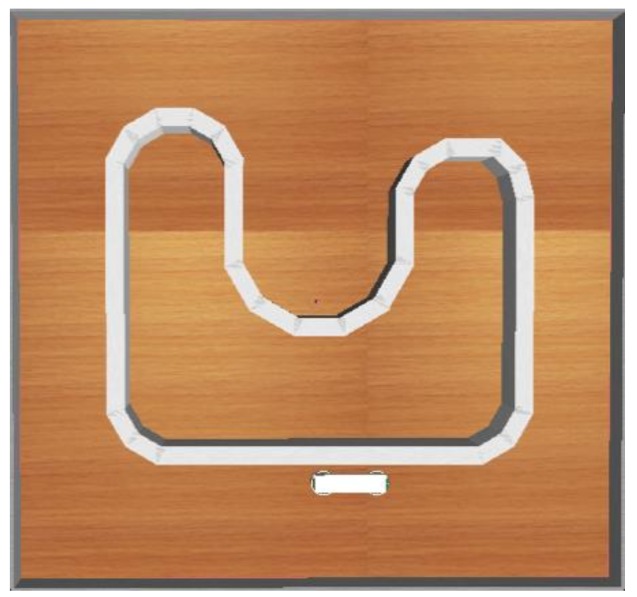
Training environment for cooperative load-carrying wall-following control.

**Figure 15 sensors-18-04181-f015:**
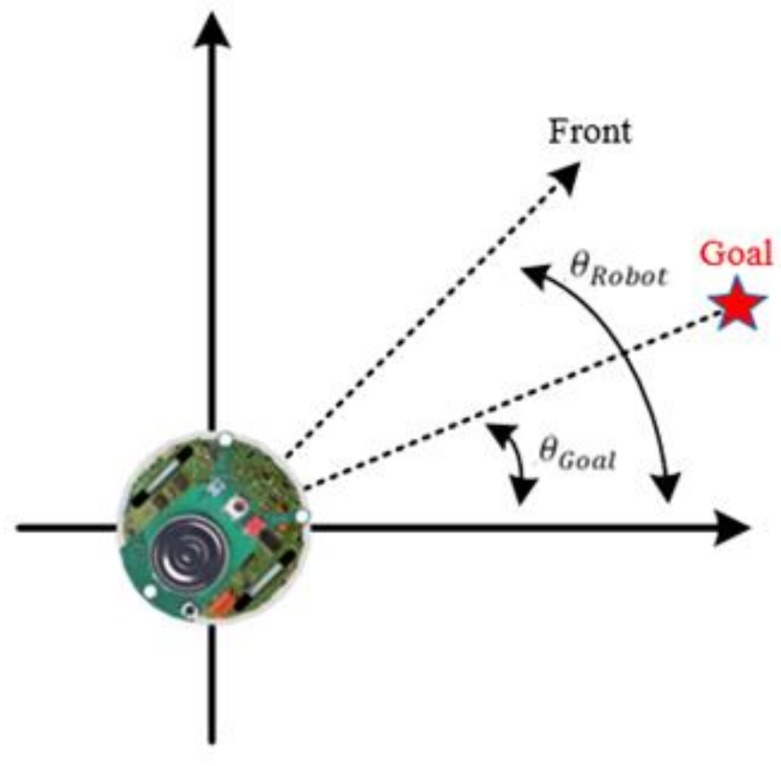
Angle $\theta_{RG}$.

**Figure 16 sensors-18-04181-f016:**
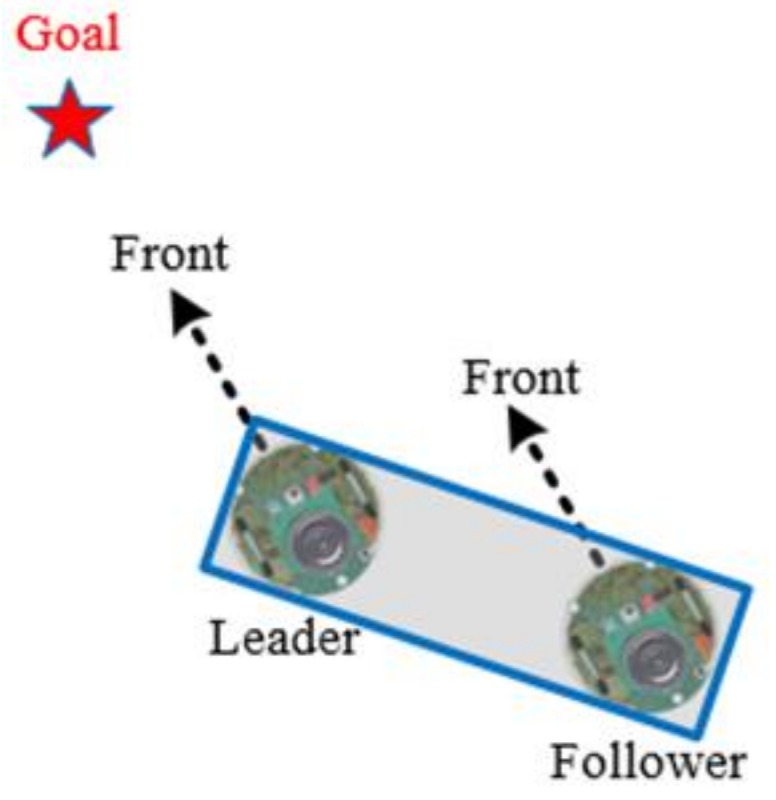
GOM.

**Figure 17 sensors-18-04181-f017:**
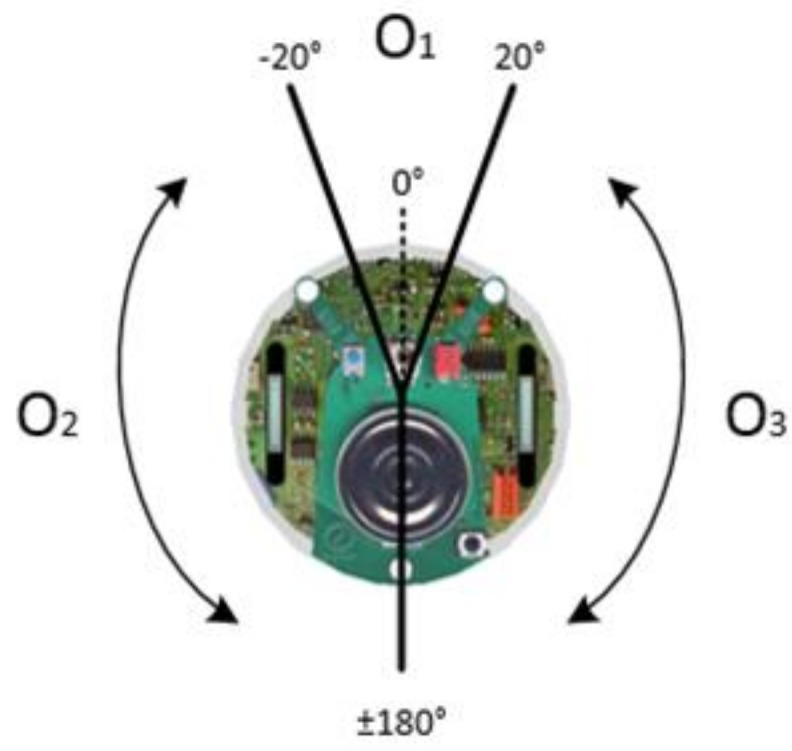
The area distribution of a robot.

**Figure 18 sensors-18-04181-f018:**
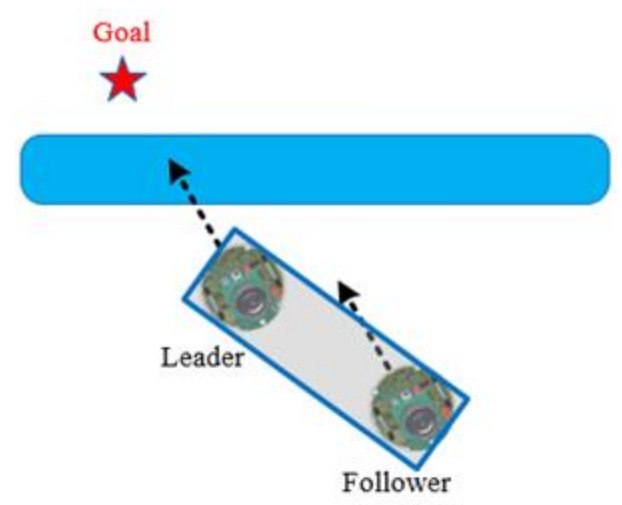
Leader detects the obstacle.

**Figure 19 sensors-18-04181-f019:**
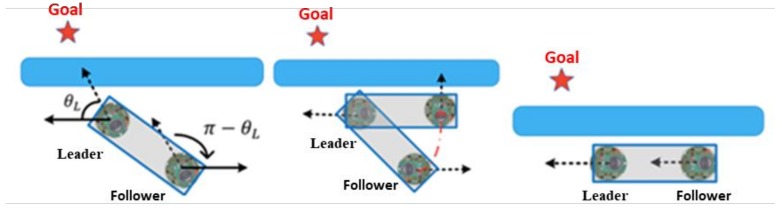
Actions of the follower.

**Figure 20 sensors-18-04181-f020:**
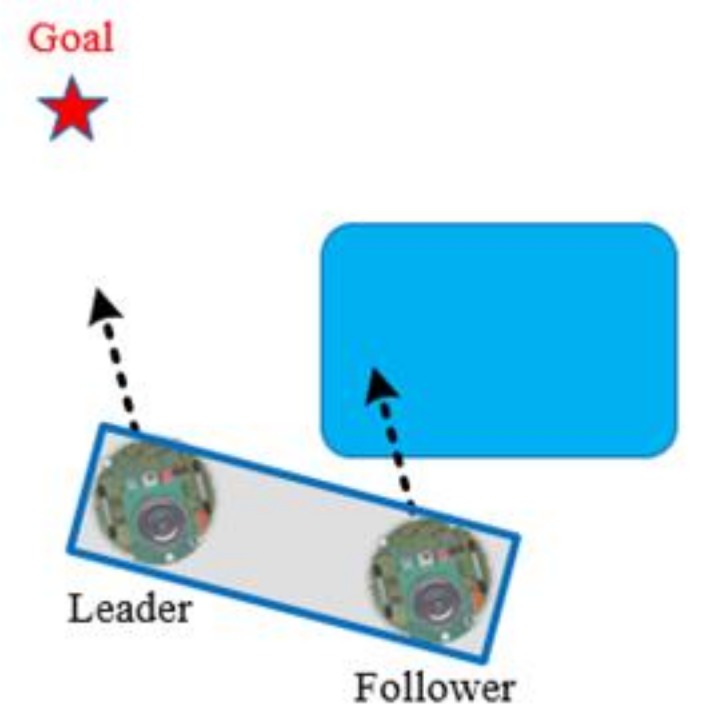
Follower detects the obstacle.

**Figure 21 sensors-18-04181-f021:**
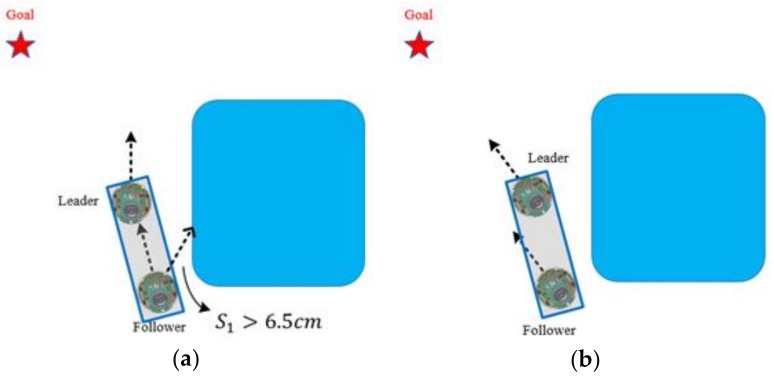
(**a**) Distance between follower and obstacle is larger than 6.5 cm; (**b**) Robots switch the WFM back to the GOM.

**Figure 22 sensors-18-04181-f022:**
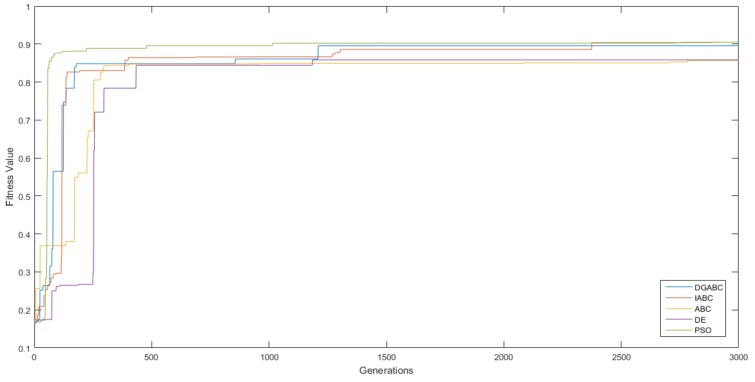
Learning curves of wall-following control in various algorithms.

**Figure 23 sensors-18-04181-f023:**
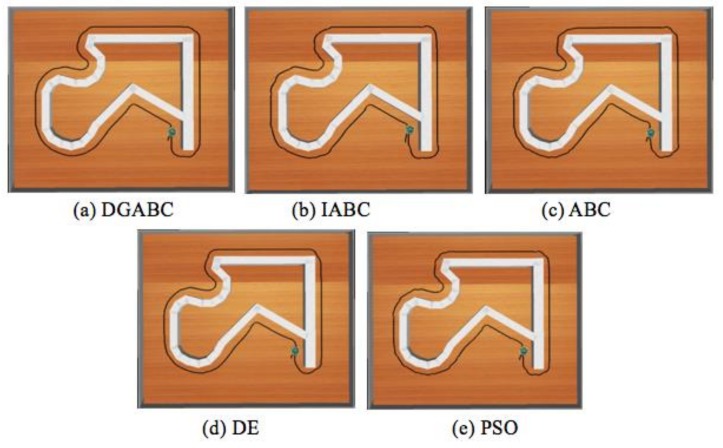
Moving path of various algorithms in the training environment.

**Figure 24 sensors-18-04181-f024:**
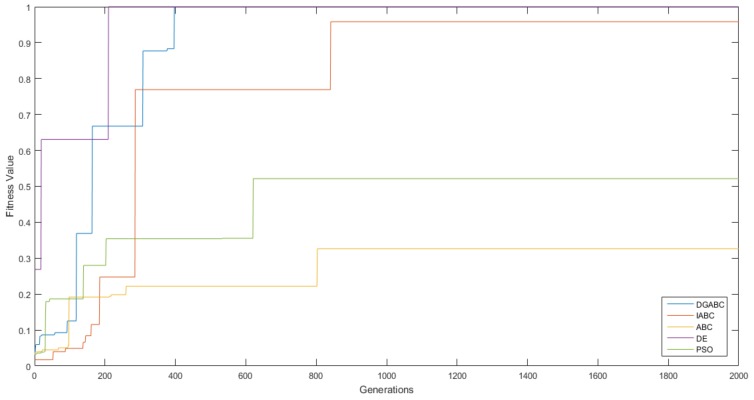
Learning curves of various algorithms.

**Figure 25 sensors-18-04181-f025:**
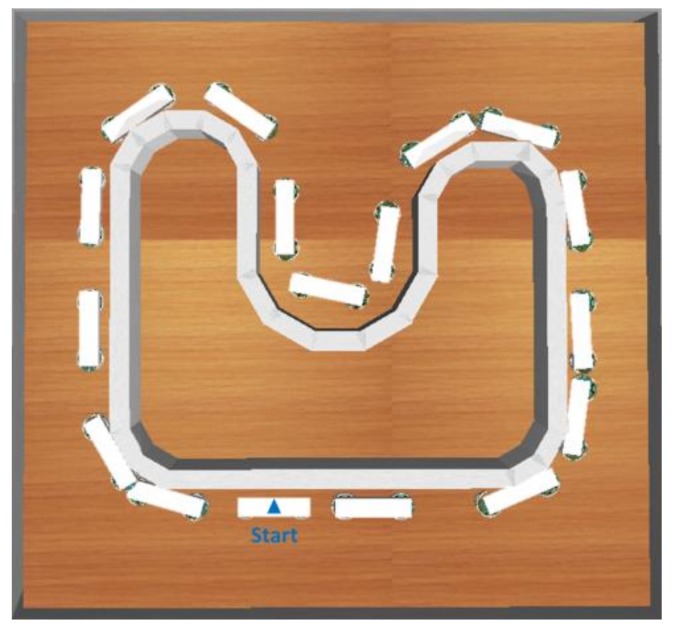
Moving path of the proposed method.

**Figure 26 sensors-18-04181-f026:**
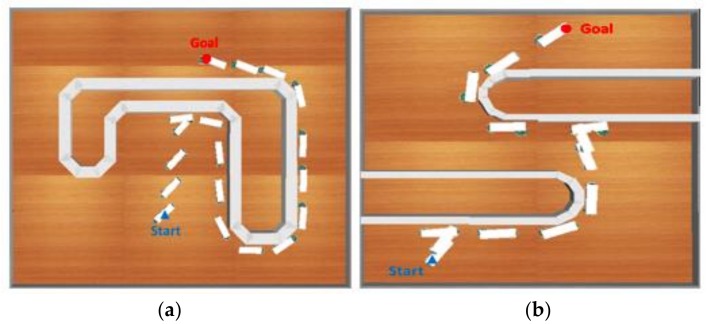
Moving path of cooperative load-carrying navigation control in (**a**) testing environment 1 and (**b**) testing environment 2.

**Figure 27 sensors-18-04181-f027:**
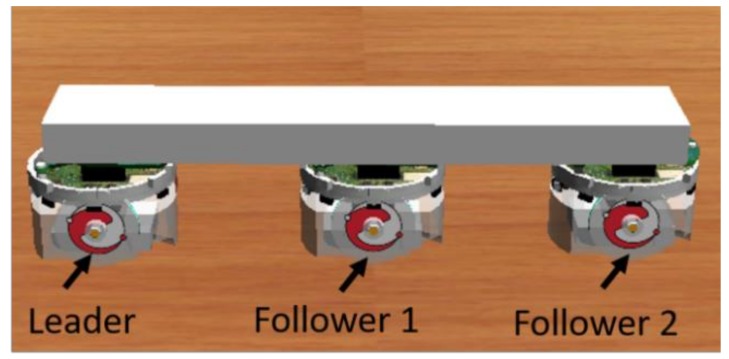
Cooperative load-carrying of three robots.

**Table 1 sensors-18-04181-t001:** Features of various robot simulator software.

Software	Platforms Supported	Dynamic Collision Avoidance	Relative End Effectors	Off-Line Programming	Real-Time Streaming Control of Hardware
Gazebo	Linux	Yes	Yes	Yes	Yes
V-Rep	Linux, mac OS, Windows	No	No	No	No
Webots	Linux, mac OS, Windows	Yes	Yes	Yes	Yes

**Table 2 sensors-18-04181-t002:** Performance comparison of various algorithms.

	Evaluation	Fitness Value				Number of Success Runs	Average Executing Times(s)
		Best	Worst	Average	STD		
Algorithm							
PSO [24]		0.912	0.905	0.908	0.003	7	227
DE [26]		0.872	0.858	0.864	0.006	8	207
ABC [27]		0.895	0.857	0.876	0.016	5	230
IABC [28]		0.926	0.905	0.917	0.009	8	199
**DGABC**		**0.895**	**0.881**	**0.889**	**0.006**	**10**	**183**

**Table 3 sensors-18-04181-t003:** Performance comparison of IT1NFC and IT2NFC.

	Evaluation	Fitness Value				Number of Success Runs	Average Executing Times (s)
		Best	Worst	Average	STD		
Algorithm							
IT1NFC (type-1)		0.889	0.856	0.878	0.011	9	198
IT2NFC (type-2)		0.895	0.881	0.889	0.006	10	183

**Table 4 sensors-18-04181-t004:** Performance of cooperative load-carrying WFM control.

	Evaluation	Fitness Value				Number of Successful Runs	Average Executing Times (s)
		Best	Worst	Average	STD		
Algorithm							
PSO [24]		0.522	0.160	0.325	0.131	5	193
DE [26]		1	0.446	0.632	0.245	6	172
ABC [27]		0.326	0.1	0.195	0.087	3	179
IABC [28]		0.958	0.247	0.583	0.250	7	165
DGABC		1	0.764	0.921	0.105	10	141

**Table 5 sensors-18-04181-t005:** Performance of cooperative load-carrying navigation control.

	Evaluation	Testing Environment 1		Testing Environment 2	
		RAD (cm)	FAD (cm)	RAD (cm)	FAD (cm)
Algorithm					
DGABC		15.64	3.72	17.08	3.81

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
