# Peer review of "Interval Type-2 Neural Fuzzy Controller-Based Navigation of Cooperative Load-Carrying Mobile Robots in Unknown Environments"

_sensors, 2018, doi:10.3390/s18124181_

Round 1

Reviewer 1 Report

The authors proposed a modified ABC algorithm for adaptively adjust an interval type-2 neurofuzzy controller for wall following and cooperative load-carrying wall following behaviours. A behaviour switching controller is presented for cooperative load carrying navigation control. Experiments were conducted to show that the proposed DGABC method outperformed other methods in adapting the interval type-2 neurofuzzy controller for robot behaviours.

The paper focuses on development of novel intelligent control algorithm for cooperative robot system but the contribution in sensors and sensing is minimal. The authors should consider submitting the manuscript to other robotics related journals.

The authors did not explain clearly why an interval type-2 neurofuzzy controller is adopted in robot behaviour learning. What is the advantages of adopting interval type-2 fuzzy logic system in the robot behaviour control?

In line 195, "FN" should read "SN".

The authors are urged to discuss the computational complexity of the proposed DGABC compared with traditional ABC.

Are the experiments for behaviour training and testing conducted in robot simulation? If that's the case, the authors are urged to conduct experiments using real robots.

The authors are urged to discuss the extensibility of the proposed method to load carrying using more than 2 robots.

Author Response

Please see attachment for more details

Reviewer 2 Report

This manuscript presents a navigation methodology for cooperative load-carrying mobile robotics. It also addresses the behavior manager that deals different modes (wall-following and goal oriented).

Authors describe the controller methods and optimizations and finally they use the well-known e-puck education robot platform to perform the tests (control the robot and Cooperative Load-Carrying task). [line 217: The -> the]

Results are composed by a complete run of trials that allow to compare with other algorithms.

The manuscript sound scientifically correct and well written.

Author Response

Please see attachment for more details

Reviewer 3 Report

really nice piece of work, and well presented, you need to modify the following:

-Refer inside the text to the references in a serial order of appearance and not in a mixed up order, like in lines 52-53.

-Include in your reference list more than the half of them (as is now) reports+papers from the last 5 years, given the fact that technology in the domain of robotics is rapid progressed. you could try papers like, 

M. Papoutsidakis, K. Kalovrektis, C. Drosos and G. Stamoulis, ‘Design of an Autonomous Robotic Vehicle for Area Mapping and Remote Monitoring’, International Journal of Computer Applications, (ISSN: 0975 – 8887), Vol. 167, No 167, June 2017

Author Response

Please see attachment for more details

Round 2

Reviewer 1 Report

I appreciate the attempt of the authors in putting more emphasis on sensing aspect of the proposed work, but many points still need clarification and revision.

In line 54, "sensor inaccuracy" is a type of "sensor uncertainty" and so "sensor inaccuracy" is redundant in this sentence.

In line 55-56, what specific aspects of sensor uncertainty considered in the proposed cooperative navigation method for load-carrying mobile robots can only be handled by interval type 2 fuzzy sets, but not type 1 fuzzy sets? Why type 2 fuzzy set is a good choice in modelling the aspects of sensor uncertainty specific to your application (cooperative navigation of load carrying robots)? Have you considered other uncertainty modelling methods, like rough sets, soft sets, evidence theory, probability, etc.? 

In line 59, how do interval type 2 fuzzy sets "increase noise suppression of sensor and make sensor more accurate and reliable". The fuzzy sets can capture or model uncertainty involved in sensor data. I don't see how noise suppression and accuracy improvement can be achieved. Is there any update rules to minimize, for example the interval of uncertainty mean of the Gaussian membership function and the standard deviation of type 2 fuzzy sets in your algorithm?

In line 62, ".. for reducing the rank", what rank is reduced?

The advantages of using type 2 neurofuzzy controller in robot behaviour control are not clearly explained. In "Reply to Comment 2", the authors only plainly show the major components of the proposed method, like WFM, GOM, DGABC and so on.

In Reply to Comment 5, Saying that the simulation software "is used in more than 1200 universities and industrial research centers worldwide" cannot provide solid support for testing the proposed algorithm in simulation. What features does the Webot simulator have in order to perform realistic simulation? Why Webot? Have you tried other simulators, like Gazebo, V-REP and so on?

Round 3

Reviewer 1 Report

I appreciate that the authors provide justification of using Interval Type 2 fuzzy sets in modelling sensor uncertainty in the proposed controller.

Change all "et al." to italic as it is Latin.

Include simulation result of Type 1 fuzzy logic controller in the manuscript to justify the use of Type 2 fuzzy logic controller in the paper.

Why Webot is adopted in simulation? Why not other simulators like V-REP, Gazebo, etc.?

Author Response

Attached please find the revised manuscript
